

# Distribution and host range of viruses associated with the citrus leprosis disease complex in Mexico

Hugo Enrique González-García[1], Ma. Teresa Santillán-Galicia[1], Laura Delia Ortega-Arenas[1], Guadalupe Valdovinos-Ponce[2], Alberto Enrique Becerril-Román[3], Pedro Luis Robles-García[4], Ariel Wilbert Guzmán-Franco[1] and Alfredo Sánchez-Villarreal[5]

[1] Fitosanidad-Entomología y Acarología, Colegio de Postgraduados, Texcoco, Estado de Mexico, Mexico
[2] Fitosanidad-Fitopatología, Colegio de Postgraduados, Texcoco, Estado de Mexico, Mexico
[3] Fruticultura, Colegio de Postgraduados, Texcoco, Estado de Mexico, Mexico
[4] Campañas de Prioridad Nacional, Dirección General de Sanidad Vegetal, CDMX, CDMX, Mexico
[5] Campus Campeche, Colegio de Postgraduados, Champotón, Campeche, Mexico

## ABSTRACT

Citrus leprosis virus C (CiLV-C) (*Cilevirus*) and orchid fleck virus citrus strain (OFV-Cit) (*Dichorhavirus*) are viruses associated with citrus leprosis disease. Although symptoms associated with CiLV-C were observed in orange in 2005 in Mexico, and confirmed using molecular techniques in 2011, no studies have been made on the distribution of either CiLV-C or OFV-Cit viruses. During 2017, we studied the geographical distribution and host range of these two viruses infecting citrus orchards in Mexico, specifically orange, lime, mandarin and grapefruit orchards in 15 Mexican states. Furthermore, in 2019 we sampled lime orchards in three Mexican states. Presence of CiLV-C and OFV-Cit was determined using reverse transcription polymerase chain reaction (RT-PCR) assays. During 2017 the proportion of leaves infected by either CiLV-C or OFV-Cit was significantly affected by geographical origin. However, only a few samples were obtained from mandarin and grapefruit so these data were excluded from statistical analysis; orange had significantly higher rates of infection with CiLV-C than lime and the opposite was observed for OFV-Cit. Using RT-PCR, some asymptomatic leaves from 2017 samples were positive for the viruses of interest. In 2019 no symptoms associated with leprosis were observed in any of the leaves sampled from lime orchards. However, low infection rates were detected, with 6% of samples testing positive for CiLV-C and 3% for OFV-Cit. To confirm the identity of the CiLV-C isolate found in lime leaves collected in 2019, we sequenced nearly the complete RNA2 genomic region of the virus. Basic Local Alignment Search Tool (BLAST) search revealed 98.99% similarity with previously reported CiLV-C sequences from other citrus species. The implications of our results for field monitoring and disease detection are discussed.

Corresponding author
Ma. Teresa Santillán-Galicia,
teresa.santillan.galicia@gmail.com

## INTRODUCTION

Citrus production is one of the most economically significant agricultural activities in the world. In 2023, production of oranges was estimated at 70 million tons, making them the fifth most produced fruit worldwide after bananas, watermelons, apples and grapes (*Statista, 2024*). Viral diseases can affect the productivity and health of citrus trees significantly; citrus leprosis (CL) disease, caused by a complex of viruses, is one of the most economically important and mainly affects orange (*Citrus × sinensis* (L.) Osbeck) and mandarin (*Citrus reticulata* Blanco) (*Bastianel et al., 2006*). Mite species within the genus *Brevipalpus* are vectors of the viruses that cause this disease (*Rodrigues et al., 2000*; *Garita et al., 2014*; *Leon et al., 2017*; *García-Escamilla et al., 2018*). Currently, CL occurs in most Southern and all Central American countries and, since 2005, has also been recorded in Mexico (*Izquierdo-Castillo et al., 2011*; *Roy et al., 2015*).

Viruses associated with CL include cytoplasmatic-type viruses from the genera *Cilevirus* and *Higrevirus*, and nuclear type viruses from the genus *Dichorhavirus*. Cytoplasmic viruses in the genus *Cilevirus* include citrus leprosis virus C (CiLV-C) (*Locali-Fabris et al., 2006*) and citrus leprosis virus C2 (CiLV-C2) (*Roy et al., 2013*); and from the genus *Higrevirus*, hibiscus green spot virus 2 (HGSV-2) (*Melzer et al., 2012*). Nuclear type viruses include: citrus leprosis virus N (CiLV-N) (*Ramos-González et al., 2017*); the viruses reported by *Roy et al. (2015)* and *Cruz-Jaramillo et al. (2014)*, which are considered as citrus strains of the orchid fleck virus (OFV-Cit) (*Afonso et al., 2016*); and more recently, the citrus chlorotic spot virus (CiCSV) (*Chabi-Jesus et al., 2018*; *Amarasinghe et al., 2019*). Characteristic symptoms of CL in oranges include chlorotic or necrotic lesions on branches, leaves and fruits that the commercial value of fruits (*Bastianel et al., 2010*). Mexico produces several citrus species including orange, mandarin, lime (*Citrus × latifolia* (Yu. Tanaka) Tanaka), key lime (*Citrus × aurantifolia* [Christm.] Swingle) and grapefruit (*Citrus × paradisi* Macfad). *Salinas-Vargas et al. (2016)* reported the presence of *Brevipalpus yothersi* (Baker) and *Brevipalpus californicus* (Banks) in Mexican citrus orchards, where the former species was the most abundant and found on all citrus species and in all citrus-growing regions evaluated. In contrast, *B. californicus* was predominantly found on orange and mandarin. More recently, *Beltran-Beltran et al. (2020)* confirmed that *B. yothersi* was more predominant than *B. californicus* in citrus orchards, although these authors found more *B. californicus* on lime than on orange or sweet lime. In other countries, CL has been reported mainly in sweet citrus species like orange and mandarin, whereas limes are considered immune to CiLV-C (*Bastianel et al., 2006*). We consider that it is important to accurately determine whether leprosis viruses are present in all these citrus species under Mexican field conditions. Current CL reports often rely on symptoms, which can vary or even be absent in citrus species other than orange, leading to an underestimation of the distribution of CL. This underlines the importance of using molecular techniques for virus detection.

Mexico ranks as the world's third-largest producer of oranges and second-largest producer of limes (*The World Ranking, 2023*). Therefore, it is important to determine the status of the viruses associated with citrus leprosis disease on the key citrus species

grown in Mexico. This study aimed to determine the presence of CiLV-C (*Cilevirus*), OFV (*Dichorhavirus* type member), and two citrus strains of the orchid fleck virus (OFV-Cit) in Mexican orchards, while examining potential virus-host-geography relationships. In 2017, we conducted sampling in 15 Mexican states, targeting the most economically important citrus species: orange, mandarin, lime and grapefruit. In 2019, sampling focused exclusively on lime orchards in three states. Virus detection was achieved using reverse transcription polymerase chain reaction (RT-PCR).

## MATERIALS & METHODS

### Sampling sites and data collection

The National Service for Agrifood Health, Safety, and Quality (SENASICA) in Mexico has implemented surveillance activities for early detection of citrus leprosis through the Phytosanitary Epidemiological Surveillance Program since 2011. In accordance with this initiative, fieldwork for the present study was conducted under the supervision and coordination of SENASICA personnel. Site selection for biological sampling was made in agreement with landowners and guided by SENASICA representatives. Some sections of the 'Materials and Methods', as well as of the 'Results' and 'Discussion', were previously published as part of the first author's Master's thesis (*González-García, 2018*).

In 2017, citrus crops (orange, grapefruit, mandarin, lime) were sampled from orchards in 15 states: Campeche, Chiapas, Hidalgo, Jalisco, Morelos, Nuevo León, Oaxaca, Puebla, Querétaro, San Luis Potosí, Sinaloa, Tamaulipas, Veracruz, Quintana Roo, and Zacatecas (Fig. 1), in close collaboration with the Crop Protection Committee of each state. In each state, five orchards were sampled, except for Chiapas and Tamaulipas where seven and two orchards were sampled, respectively. The sampling methodology was the same for all citrus species and locations. In each orchard, five trees were sampled from five points: the four corners and the central part of the orchard (25 trees in total). From each tree, 20 leaves were randomly sampled, five from each cardinal point and 1.5 m above the ground. The collected leaves were deposited in plastic bags, labelled, placed inside cool boxes and transported to the laboratory. Once in the laboratory, all leaves were inspected under a stereomicroscope to identify the presence of symptoms associated with citrus leprosis disease. All leaves were then washed in sterile distilled water and incubated at room temperature for 15 min to remove excess moisture. From the 20 leaves per tree, we selected five leaves per tree for molecular detection of viruses. Selection was based on the presence of symptoms (Fig. 2), or randomly when no symptoms were evident. From each selected leaf, two circular sections of one cm diameter, each containing a lesion associated with citrus leprosis, were excised using a metal cork-borer; when no lesions were evident, two circular sections of one cm diameter were excised from the central part of the leaf, one on each side of the midrib. All sections were deposited individually into two mL Eppendorf tubes containing one mL of RNAlater™ (Sigma-Aldrich, St. Louis, MO, USA), and stored at −20 °C until required.

In 2019, and using the same methodology, we sampled only lime orchards in the states of Colima, Puebla and Sinaloa. Here we sampled two species, *Citrus × latifolia* and *Citrus*

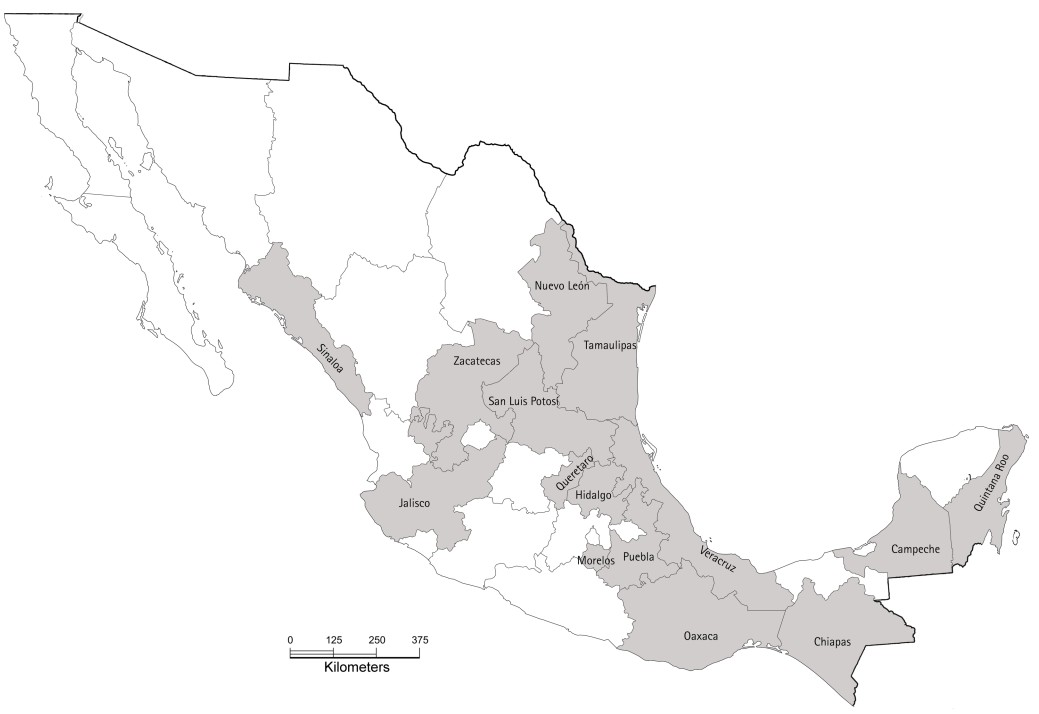

**Figure 1** **Map of Mexico showing the 15 states sampled in our study during 2017.** In each state, between two and seven orchards were sampled. The base map was obtained from INEGI's "Imprime tu mapa" tool (https://cuentame.inegi.org.mx/imprime_tu_mapa/) and edited using GIMP to highlight sampled estates. The scale bar is in kilometers.

× *aurantofolia*. In Colima, five lime orchards in different locations were sampled, five orchards in Puebla and three in Sinaloa. In all sampled sites, no evident symptoms of CL were found in any of the leaves; therefore, for the molecular detection, 10% of the leaves collected per location was randomly selected, ranging from 10 to 50 leaves depending on the size of the orchard, and processed as described above. As in 2017, sampling was done in collaboration with the Crop Protection Committee of each state.

## Molecular analysis
### RNA extraction
For RNA extraction, Eppendorf tubes containing 100 mg of leaf tissue were submerged in liquid nitrogen for 15 min. The frozen tissue was ground using a pellet pestle rod (Daigger and Company Inc., Vernon Hills, IL, USA) and RNA was extracted using the RNeasy Plant Mini Kit (QIAGEN GmbH, Hilden, Germany) following the manufacturer's instructions. The concentration of RNA in the samples was estimated using a NanoDrop™ and stored at −20 °C until required.

### RT-PCR assays, sequencing and analysis
Virus detection was achieved by reverse transcription polymerase chain reaction (RT-PCR). Reverse transcriptase reactions were done using the RevertAid™ First Strand cDNA Synthesis Kit (Thermo Fisher Scientific Inc. Waltham, MA, USA). Each RT reaction was

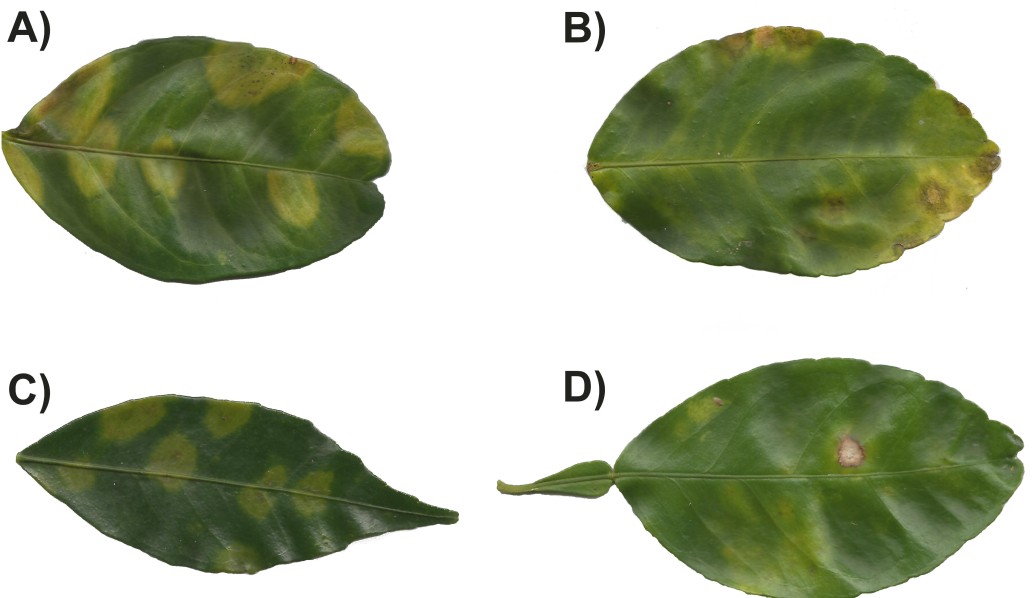

**Figure 2  Symptoms observed in leaves with citrus leprosis from different citrus species.** (A) *Citrus × sinensis*, (B) *Citrus × latifolia*, (C) *Citrus reticulata*, and (D) *Citrus × paradise*.

done in a final volume of 20 µL containing four µL of 5X reaction buffer, one mM of each dNTP, five µM of random hexamer primer, one µL of RevertAid M-MuLV RT enzyme (200 U/µL), 10 µL nuclease-free water and two µL of RNA template (approx. 80 ng). Tubes were incubated at 25 °C for five min, and then 60 min at 42 °C followed by a final step of 5 min at 70 °C in a MyCycler™ thermal cycler (BIO-RAD Laboratories Inc., Hercules, CA, USA).

Presence of the viruses CiLV-C, OFV and OFV-Cit were evaluated in each leaf sample. Detection was achieved using PCR assays with complementary DNA (cDNA) from the RT reactions previously described, as a template. CiLV-C was detected using the primers MPF and MPR (*Locali et al., 2003*). OFV was detected using the primers OFVF-OFVR (*Ali et al., 2014*). OFV-Cit was detected using the primer sets NPF-NPR (*Roy et al., 2015*) and CNSV2F-CNSV2R (*Cruz-Jaramillo et al., 2014*). These two strains of OFV will be referred to hereafter as OFV-Cit1 and OFV-Cit2, respectively. For all pairs of primers, PCR assays were done in a final volume of 25 µL containing 2.5 µL of 10X PCR buffer (Tris-Cl, KCl, (NH4)2SO4, 15 mM MgCl2; pH 8.7), 0.18 µM of each primer, 0.2 mM of each dNTP, 0.2 µL of Taq polymerase (five U/µL) (QIAGEN, GmbH, Hilden, Germany) and three µL of cDNA. Negative controls (nuclease-free water replacing template) were included in every PCR run to monitor for contamination. Thermal conditions for each primer pair were as follows: CiLVC—one cycle of 5 min at 94 °C, 35 cycles of 1 min at 94 °C, 1 min at 57 °C and 1 min a 72 °C, followed by a final extension of 72 °C for 10 min. OFV and OFV-Cit1—one cycle of 5 min at 95 °C, 35 cycles of 1 min at 95 °C, 1 min at 61 °C and 1 min at 72 °C, followed by a final extension of 72 °C for 10 min; and OFV-Cit2—one cycle of 5 min at 94 °C, 35 cycles of 35 s at 94 °C, 20 s at 62 °C and 45 s at 72 °C, followed by a final extension

of 72 °C for 10 min. All PCR assays were done in a MyCycler™ thermal cycler. PCR products were visualized on 1.5% agarose gels in 1X TAE. Gels were stained with ethidium bromide (10 mg mL$^{-1}$) and photographed. Expected band sizes were 339 bp for CiLV-C, 160 bp for OFV, 681 bp for OFV-Cit1 and 480 bp for OFV-Cit2. Although the presence of these viruses in plant tissues was demonstrated by the size of the PCR products present, we also sent some PCR products to Macrogen Inc. (Geumchen-gu, Seoul, Korea) for direct sequencing to confirm the identity of the viruses. Resulting sequences were edited using BioEdit v.7.1.9 (*Hall, 1999*). Multiple alignments were made using Clustal W (*Thompson, Higgins & Gibson, 1994*) implemented in BioEdit. After alignment and trimming, the final lengths of the sequences were 339, 105, 670 and 473 for CiLV-C (17 sequences), OFV (three sequences), OFV-Cit1 (six sequences) and OFV-Cit2 (five sequences), respectively. The Basic Local Alignment Tool (BLAST) implemented in GenBank was used to confirm the identity of the sequences belonging to each of the viruses found. GenBank accession numbers are shown in Table 1.

To confirm the identity of CiLV-C detected in lime leaves (*Citrus × latifolia*) in 2019, we sequenced nearly the entire RNA2 genomic region of the virus found collected in 2019. For this, we designed five pairs of internal primers based on the complete RNA2 genome sequence of CiLV-C (GenBank: NC_008170), spanning the entire genomic segment. The primers used are listed in Table S1. The obtained sequence is available at GenBank (PV533983).

### Data analysis

The presence per leaf of individual viruses and combinations of two viruses at the same time were recorded for each citrus species and geographic location in leaves with symptoms (Table 2) and without symptoms (Table 3) collected in 2017. A series of logistic regressions were done to analyze infection data for CiLV-C and the combined data for OFV, OFV-Cit1 and OFV-Cit2 (OFV/OFV-Cit) separately. Data for mandarin and grapefruit were excluded from the analysis because of the low number of resulting samples (Table 2). For each virus data set (*i.e.,* CiLV-C and OFV/OFV-Cit), the proportion of infected leaves were compared (considering a binomial distribution with sample size equal to the total number of leaves analyzed), using hierarchical contrasts where the effect of citrus species (lime or orange) was compared first, then, within each citrus species, the effect of geographical origin was assessed. Only treatments with a sample size above 15 were analyzed. The final sample sizes for the different treatments varied between 15 and 25. All analyses were made using GenStat v 24.0 (*VSN International, 2022*). Data obtained from sampling carried out in 2019 were not analyzed due to the low proportion of leaves positive for the virus (Table 4).

## RESULTS

All viruses studied were detected in all citrus species regardless of symptom presence (Tables 2 and 3). Dual infections by CiLV-C and OFV/OFV-Cit were most frequent in lime (approx. 15%) and less common in orange (approx. 7%) (Tables 2 and 3).

CiLV-C infection rates differed significantly between lime and orange ($\chi_1^2 = 7.37$, $P = 0.007$), with a higher proportion observed in orange leaves compared with lime

**Table 1  GenBank accession numbers.** Virus samples used for sequencing with their plant host and geographical origin.

| Code | Plant host | State | GenBank |
|---|---|---|---|
| | | CiLV-C | |
| C1 | Lime | Campeche | MT056048 |
| C3 | Mandarin | Campeche | MT056050 |
| C5 | Lime | Campeche | MT056051 |
| C6 | Lime | Campeche | MT056052 |
| CH1 | Orange | Chiapas | MT056053 |
| CH2 | Orange | Chiapas | MT056054 |
| CH3 | Orange | Chiapas | MT056055 |
| CH4 | Orange | Chiapas | MT056056 |
| J3 | Lime | Jalisco | MT056058 |
| J4 | Lime | Jalisco | MT056059 |
| N1 | Grapefruit | Nuevo Leon | MT056060 |
| N2 | Grapefruit | Nuevo Leon | MT056061 |
| N4 | Orange | Nuevo Leon | MT056062 |
| N6 | Grapefruit | Nuevo Leon | MT056063 |
| N7 | Orange | Nuevo Leon | MT056064 |
| Q2 | Orange | Queretaro | MT056065 |
| QR2 | Orange | Quintana Roo | MT056066 |
| QR3 | Orange | Quintana Roo | MT056067 |
| V5 | Orange | Veracruz | MT056068 |
| Z1 | Lime | Zacatecas | MT056069 |
| Z3 | Lime | Zacatecas | MT056057 |
| Z6 | Lime | Zacatecas | MT056049 |
| | | OFV | |
| C2 | Lime | Campeche | MT073364 |
| J1 | Lime | Jalisco | MT073365 |
| N5 | Grapefruit | Nuevo Leon | MT073366 |
| V1 | Orange | Veracruz | MT073368 |
| Z10 | Lime | Zacatecas | MT073367 |
| | | OFV-Cit1 | |
| Z7 | Orange | Zacatecas | MT073361 |
| QR1 | Orange | Quintana Roo | MT073362 |
| V2 | Orange | Veracruz | MT073363 |
| | | OFV-Cit2 | |
| N8 | Orange | Nuevo Leon | MT073355 |
| Z5 | Lime | Zacatecas | MT073356 |
| Z8 | Orange | Zacatecas | MT073357 |
| S1 | Orange | San Luis Potosi | MT073358 |
| S2 | Orange | San Luis Potosi | MT073359 |
| S3 | Orange | San Luis Potosi | MT073360 |

**Table 2  Citrus leprosis viruses detected in leaves with symptoms that were sampled from citrus species from different Mexican states during 2017.** Numbers in columns represent the number of leaf samples that were positive for each of the viruses evaluated, either alone or as a dual infection. The lime species was *Citrus × latifolia* for all samples.

| State | Host | Leaves analysed | CiLV-C | OFV/OFV-Cit | | | CiLV-C + OFV | CiLV-C + OFV-Cit2 | OFV + OFV-Cit2 | OFV-Cit1 + OFV-Cit2 | No virus |
|---|---|---|---|---|---|---|---|---|---|---|---|
| | | | | OFV | OFV-Cit1 | OFV-Cit2 | | | | | |
| | Lime | 10 | 5 | 0 | 0 | 0 | 3 | 0 | 0 | 0 | 2 |
| Campeche | Mandarin | 5 | 1 | 0 | 0 | 0 | 2 | 0 | 0 | 0 | 2 |
| | Orange | 5 | 2 | 0 | 0 | 0 | 0 | 0 | 0 | 0 | 3 |
| Chiapas | Lime | 2 | 0 | 0 | 0 | 0 | 0 | 0 | 1 | 0 | 1 |
| | Orange | 8 | 4 | 0 | 0 | 0 | 0 | 0 | 0 | 0 | 4 |
| Hidalgo | Orange | 9 | 2 | 0 | 0 | 0 | 0 | 0 | 1 | 0 | 6 |
| Jalisco | Lime | 25 | 4 | 6 | 0 | 0 | 0 | 0 | 0 | 0 | 15 |
| Morelos | Lime | 14 | 1 | 2 | 0 | 1 | 0 | 0 | 0 | 0 | 10 |
| | Orange | 5 | 0 | 1 | 0 | 0 | 0 | 0 | 1 | 0 | 3 |
| Nuevo Leon | Orange | 11 | 9 | 0 | 0 | 1 | 0 | 1 | 0 | 0 | 0 |
| | Grapefruit | 2 | 0 | 0 | 0 | 0 | 0 | 1 | 0 | 0 | 1 |
| Oaxaca | Lime | 18 | 1 | 2 | 0 | 0 | 1 | 1 | 1 | 0 | 12 |
| | Orange | 3 | 0 | 2 | 0 | 0 | 0 | 0 | 0 | 0 | 1 |
| Puebla | Lime | 10 | 0 | 3 | 0 | 0 | 0 | 0 | 3 | 0 | 4 |
| Queretaro | Orange | 24 | 8 | 0 | 0 | 0 | 0 | 0 | 4 | 0 | 12 |
| Quintana Roo | Orange | 24 | 9 | 0 | 1 | 0 | 0 | 0 | 0 | 0 | 14 |
| San Luis Potosi | Orange | 24 | 4 | 2 | 0 | 2 | 0 | 0 | 1 | 0 | 15 |
| Sinaloa | Orange | 5 | 3 | 0 | 0 | 1 | 0 | 0 | 0 | 0 | 1 |
| | Grapefruit | 3 | 0 | 1 | 0 | 0 | 0 | 0 | 0 | 0 | 2 |
| Tamaulipas | Orange | 23 | 10 | 1 | 0 | 0 | 4 | 0 | 1 | 0 | 7 |
| Veracruz | Orange | 20 | 9 | 0 | 0 | 0 | 0 | 0 | 0 | 0 | 11 |
| | Lime | 11 | 0 | 0 | 0 | 2 | 1 | 2 | 0 | 1 | 5 |
| Zacatecas | Grapefruit | 5 | 2 | 1 | 0 | 1 | 0 | 0 | 0 | 0 | 1 |
| | Orange | 4 | 1 | 0 | 0 | 2 | 0 | 0 | 0 | 0 | 1 |
| **Total leaves** | | 270 | 75 | 21 | 1 | 10 | 11 | 5 | 13 | 1 | 133 |

(Fig. 3). Amongst lime-producing states, infection rates varied significantly ($\chi_5^2 = 17.85$, $P = 0.003$), with the highest rate in Campeche (approx. 6%), while other states remained below 2% (Fig. 4A). A similar trend was observed in oranges, with significant variation among states ($\chi_7^2 = 23.84$, $P = 0.001$); Nuevo Leon showed the highest infection rate (approx. 8%), with other states reaching 5% (Fig. 4B).

OFV/OFV-Cit infection rates were also more prevalent in lime than orange ($\chi_1^2 = 7.01$, $P = 0.008$) (Fig. 3). Infection rates among lime-producing states differed ($\chi_5^2 = 19.58$, $P = 0.001$), with Puebla reaching over 10%, while others remained under 5% (Fig. 4A). In contrast, no significant differences were found among orange-producing states ($\chi_7^2 = 5.57$, $P = 0.590$), and infection levels never exceeded 2% (Fig. 4B).

In the sampling conducted in 2019, out of the 270 leaves analysed, 18 were positive for CiLV-C, nine for OFV and none for OFV-Cit (Table 4). Sequencing of the RNA2 region from a CiLV-C isolate in lime yielded a 4,533 bp sequence. A subsequent BLAST search

**Table 3 Citrus leprosis viruses detected in leaves without symptoms that had been sampled from citrus species from different Mexican states during 2017.** Numbers in columns represent the number of leaf samples that were positive for each of the viruses evaluated, either alone or as a dual infection. The lime species was *Citrus × latifolia* in all samples.

| State | Host | Leaves analysed | CiLV-C | OFV/OFV-Cit | | | CiLV-C + OFV | CiLV-C + OFV-Cit2 | OFV + OFV-Cit2 | OFV-Cit1 + OFV-Cit2 | No virus |
|---|---|---|---|---|---|---|---|---|---|---|---|
| | | | | OFV | OFV-Cit1 | OFV-Cit2 | | | | | |
| Campeche | Lime | 5 | 2 | 0 | 0 | 0 | 0 | 0 | 0 | 0 | 3 |
| | Mandarin | – | – | – | – | – | – | – | – | – | 0 |
| | Orange | 2 | 1 | 0 | 0 | 0 | 0 | 0 | 0 | 0 | 1 |
| Chiapas | Lime | 1 | 0 | 0 | 0 | 0 | 0 | 0 | 0 | 0 | 1 |
| | Orange | 8 | 2 | 2 | 0 | 0 | 0 | 0 | 0 | 0 | 4 |
| Hidalgo | Orange | 16 | 0 | 3 | 0 | 0 | 0 | 0 | 0 | 0 | 13 |
| Jalisco | Lime | – | – | – | – | – | – | – | – | – | – |
| Morelos | Lime | 5 | 0 | 0 | 0 | 0 | 1 | 0 | 0 | 0 | 4 |
| | Orange | 1 | 0 | 0 | 0 | 0 | 0 | 0 | 0 | 0 | 1 |
| Nuevo Leon | Orange | 5 | 3 | 0 | 0 | 0 | 0 | 1 | 0 | 0 | 1 |
| | Grapefruit | 7 | 1 | 0 | 0 | 0 | 1 | 2 | 0 | 0 | 3 |
| Oaxaca | Lime | 2 | 0 | 0 | 0 | 0 | 1 | 0 | 0 | 0 | 1 |
| | Orange | 2 | 0 | 1 | 0 | 0 | 0 | 0 | 0 | 0 | 1 |
| Puebla | Lime | 1 | 0 | 0 | 0 | 0 | 0 | 0 | 1 | 0 | 0 |
| Queretaro | Orange | 1 | 0 | 0 | 0 | 0 | 0 | 0 | 1 | 0 | 0 |
| Quintana Roo | Orange | 1 | 0 | 0 | 0 | 0 | 0 | 0 | 0 | 0 | 1 |
| San Luis Potosi | Orange | 1 | 0 | 0 | 0 | 0 | 0 | 0 | 0 | 0 | 1 |
| Sinaloa | Orange | 2 | 0 | 0 | 0 | 0 | 0 | 0 | 0 | 0 | 2 |
| | Grapefruit | – | – | – | – | – | – | – | – | – | – |
| Tamaulipas | Orange | 2 | 0 | 0 | 0 | 0 | 0 | 0 | 0 | 0 | 2 |
| Veracruz | Orange | 5 | 0 | 1 | 1 | 0 | 0 | 1 | 0 | 0 | 2 |
| Zacatecas | Lime | 4 | 0 | 0 | 0 | 1 | 0 | 0 | 1 | 0 | 2 |
| | Grapefruit | – | – | – | – | – | – | – | – | – | – |
| | Orange | 1 | 0 | 0 | 0 | 0 | 0 | 0 | 0 | 1 | 0 |
| **Total leaves** | | 72 | 9 | 7 | 1 | 1 | 3 | 4 | 3 | 1 | 43 |

showed a maximum similarity of 98.99% with previously deposited RNA2 sequences of CiLV-C, with an *E*-value of 0.0 and 100% query coverage.

## DISCUSSION

Our results showed that the viruses we analyzed for were present in Mexican citrus orchards. Unfortunately, the lack of sufficient samples from mandarin and grapefruit did not allow a statistical analysis, as was done for orange and lime. The low number of samples of mandarin and grapefruit was because only small areas of these species are grown.

In 2017, approximately 40% of asymptomatic leaves were positive for the viruses studied (Table 3). Despite being asymptomatic, these infected leaves could represent an important source of inoculum for transmission if vectors are present in orchards, which is very possible according to *Salinas-Vargas et al. (2016)*. Current leprosis management strategies typically focus on controlling the mite vector, *B. yothersi*, based solely on observations of

**Table 4  Citrus leprosis viruses detected in leaves from two lime species during 2012.** All leaves were without symptoms. Samples were taken in three Mexican states. Numbers in columns represent the number of leaf samples that were positive for each of the viruses evaluated.

| State | Locality | Host | Leaves analysed | CiLV-C | OFV | OFV-Cit1 | OFV-Cit2 | No virus |
|---|---|---|---|---|---|---|---|---|
| | El Basurero | *Citrus × aurantifolia* | 10 | 2 | 0 | 0 | 0 | 8 |
| | R. Sante Fe | *C. aurantifolia* | 10 | 3 | 0 | 0 | 0 | 7 |
| Colima | Cerro de Ortgea | *C. aurantifolia* | 15 | 4 | 0 | 0 | 0 | 11 |
| | R. Dos Rositas | *C. aurantifolia* | 22 | 5 | 0 | 0 | 0 | 17 |
| | San Francisco | *Citris × latifolia* | 10 | 4 | 0 | 0 | 0 | 6 |
| | San Pedro | *C. latifolia* | 10 | 0 | 0 | 0 | 0 | 10 |
| | Ayotla | *C. latifolia* | 17 | 0 | 0 | 0 | 0 | 17 |
| Puebla | Tilapa | *C. latifolia* | 10 | 0 | 0 | 0 | 0 | 10 |
| | Acateno1 | *C. latifolia* | 20 | 0 | 0 | 0 | 0 | 20 |
| | Acateno2 | *C. latifolia* | 30 | 0 | 0 | 0 | 0 | 30 |
| | Mazatlan | *C. latifolia* | 16 | 0 | 0 | 0 | 0 | 16 |
| Sinaloa | Navolato | *C. latifolia* | 50 | 0 | 9 | 0 | 0 | 41 |
| | Coliacan | *C. aurantifolia* | 50 | 0 | 0 | 0 | 0 | 50 |
| **Total leaves** | | | 270 | 18 | 9 | 0 | 0 | 243 |

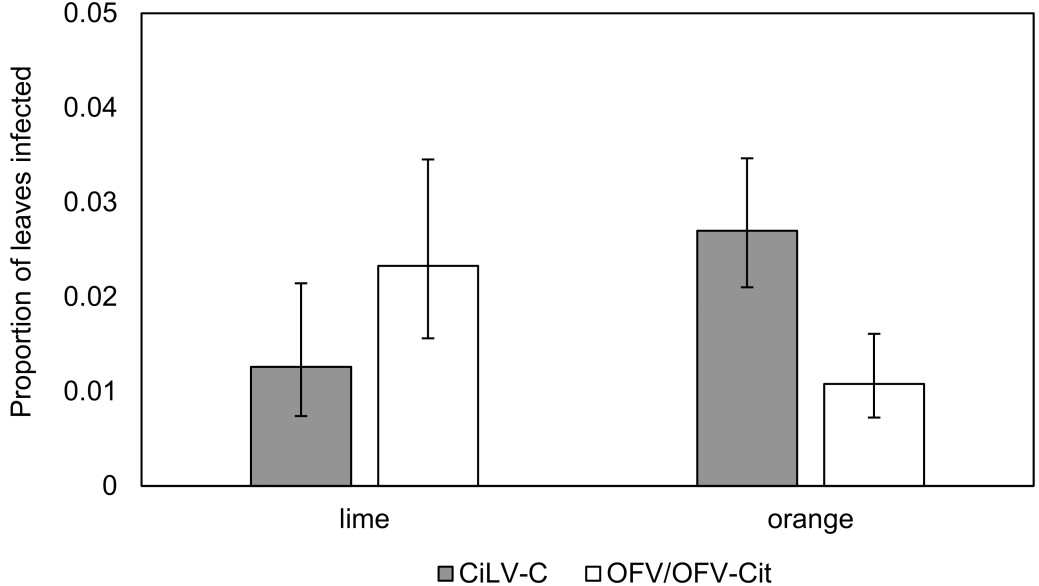

**Figure 3  Proportion of lime and orange infected leaves with CiLV-C or OFV/OFV-Cit.** Error bars represent 95% confidence intervals back transformed from the logistic scale.

evident symptoms (*Bastianel et al., 2010*). Using RT-PCR detection of virus would certainly provide more accuracy. However, because the virus is not systemic, only developing in localized areas, this approach may be impractical and may lead to an underestimation of the actual presence of the virus. Thus, detecting the virus in mites would be a more reliable approach (*Beltran-Beltran et al., 2020*).

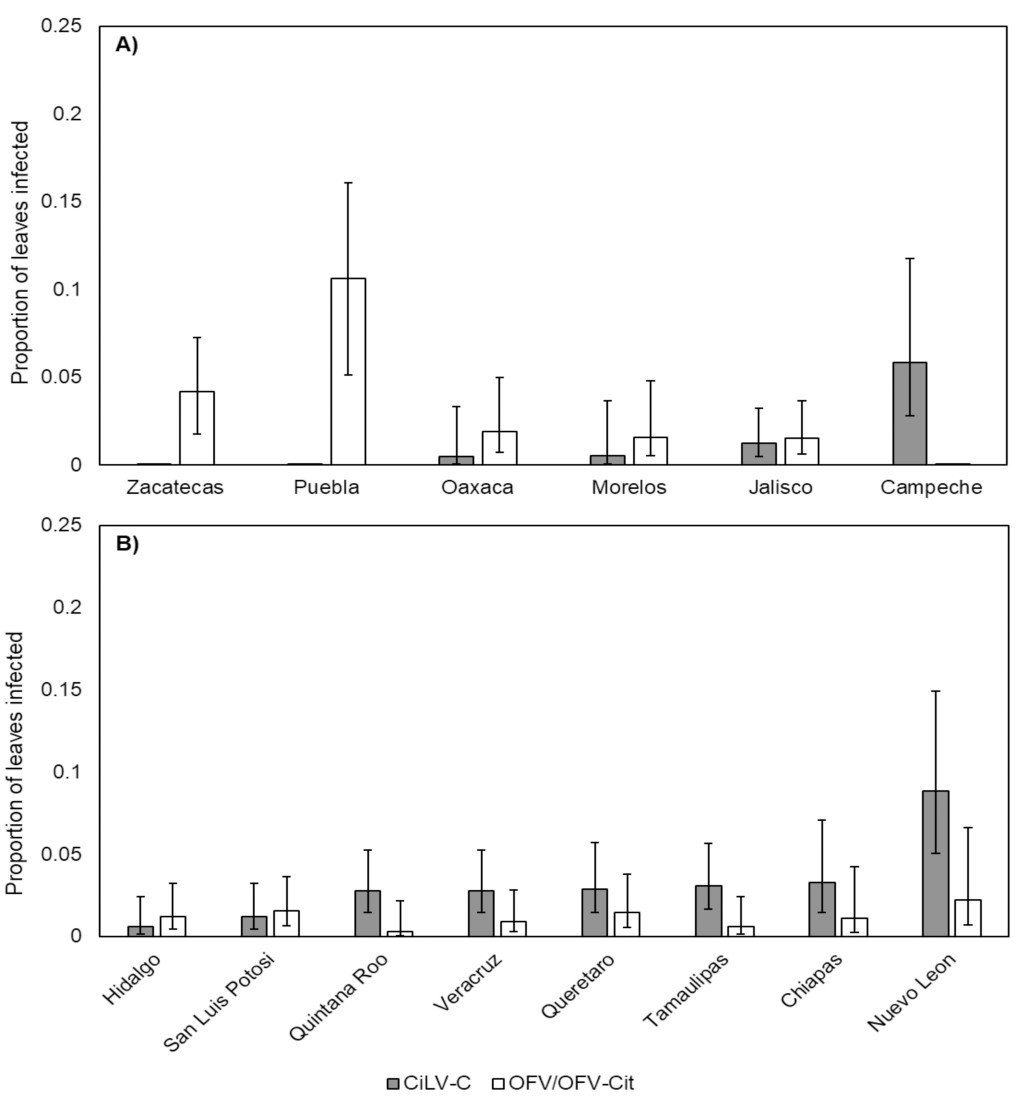

**Figure 4** **Proportion of leaves infected with CiLV-C or OFV/OFV-Cit in lime (A) and orange (B) leaves from different geographical origins (states).** Error bars represent 95% confidence intervals back transformed from the logistic scale.

    The viruses studied were detected in all citrus species, with CiLV-C being more prevalent than OFV/OFV-Cit (Tables 2, 3 and 4). The low prevalence of OFV/OFV-Cit strains was consistent across all citrus species. This was further confirmed in the 2019 sampling, which focused solely on lime orchards: only nine out of 270 leaves tested positive for OFV, and these were confined to a single location (Table 4). Dual infection of leaves was uncommon compared with infection by only one virus. This pattern may reflect viral competition or superinfection exclusion, where one virus dominates in co-infected tissues, as observed in other plant-virus systems (*Alcaide et al., 2020*). Outcomes of dual-infection by viruses causing citrus leprosis, and their interaction with vectors is still to be investigated and will provide further insight into the ecology of the virus-vector interaction in citrus orchards.

In 2017, the occurrence of CiLV-C was influenced by both citrus species and geographical origin. A greater proportion of CiLV-C infected leaves was found in orange compared with lime (Fig. 3). *Rodríguez-Ramírez et al. (2019)* reported CiLV-C-positive lime leaves that had been exposed to viruliferous *B. yothersi* adults under greenhouse conditions. The proportion of infected lime leaves was lower than that observed in orange or mandarin under the same conditions, and no symptomatology was observed during their study. Lime has been considered immune to CiLV-C infection (*Bastianel et al., 2006*). However, it is important to point out that the symptoms we observed in lime leaves during 2017, were not the typical symptoms associated with leprosis (Fig. 2B). Symptoms in orange were similar to those described by *Bastianel et al. (2006)*, whereas in lime the symptoms could be characterized as smaller with necrotic spots that had an irregular shape (Fig. 2B) (*Santillan-Galicia et al., 2022*). In lime leaves showing symptoms, it was common to find more than one virus in the same leaf (Table 2), suggesting that the symptomatology could be more closely associated with OFV/OFV-Cit infection; this needs to be confirmed experimentally. Furthermore, while we detected CiLV-C in lime leaves, their response appears to differ from the symptomatology observed in orange leaves. Symptom expression varies considerably amongst citrus species and cultivars (*Bastianel et al., 2006*), suggesting host-specific differences in tolerance. Future studies should quantify viral loads and symptom severity in lime leaves to confirm whether they are less affected than orange leaves. The absence of leprosis symptoms in lime leaves was further confirmed in samples taken in 2019 when no symptoms were observed, even though 18 out of 270 lime leaves were positive for the virus. After conducting a BLAST search using the sequence obtained from the CiLV-C isolate found in lime leaves, we confirmed that it was very similar to the CiLV-C sequences already deposited in GenBank, showing 98.99% similarity. The lack of substantial differences between our sequence and those in GenBank suggest no genetic changes that would indicate an altered host range. We believe that lime may have been considered immune largely due to absence of typical symptomology, rather than confirmed resistance.

The role of *B. yothersi* in transmission of CiLV-C has been well documented in sweet orange (*Bastianel et al., 2018*). Although transmission of CiLV-C in lime is poorly studied, *Rodríguez-Ramírez et al. (2019)* reported CiLV-C transmission in lime leaves by *B. yothersi*, but in lower proportions than in orange or mandarin. These authors also reported that the survival of *B. yothersi* on lime trees was very poor, suggesting that this could be the reason why very low proportions of lime leaves tested positive for CiLV-C. The incidence of these viruses varied amongst the states studied, but, from our current data, it is difficult to determine the specific factors affecting this. Incidence of viruses can be multifactorial, and more data are needed to infer which factors affect the prevalence of the viruses we studied; this would include monitoring disease progression over time in different citrus species and monitoring levels and sources of inoculum in relation to climatic data from the different locations.

Overall, OFV/OFV-Cit and CiLV-C were found in the same states but the proportion of both orange and lime leaves positive for OFV/OFV-Cit was low compared with the proportion of leaves positive for CiLV-C (Fig. 4). OFV-Cit transmission has been

particularly associated with *B. californicus* (*García-Escamilla et al., 2018*) rather than the most abundant mite species in Mexican citrus orchards, *B. yothersi* (*Salinas-Vargas et al., 2016*). However, similar proportions of *B. yothersi* and *B. californicus* tested positive for CiLV-C and OFV-Cit (*Beltran-Beltran et al., 2020*).

## CONCLUSIONS

In conclusion, the viruses we studied were present in all four citrus species evaluated. Prevalence of CiLV-C was highest in orange, while prevalence of OFV/OFV-Cit was highest in lime. The low detection rate of OFV/OFV-Cit and the infrequent occurrence of dual infections suggest these viruses may have restricted distribution or limited competitiveness during co-infection of citrus tissues. Our finding indicates that CiLV-C infections in lime leaves may be inapparent, highlighting the potential role of asymptomatic plant hosts in the epidemiology of citrus leprosis. We recommend that both molecular diagnostic tools and visual assessment be employed in virus detection efforts, with greater emphasis on identification of viruliferous vectors rather than solely on symptomatic citrus tissues, considering the localized and sometimes symptomless nature of these infections.

## ACKNOWLEDGEMENTS

The authors are grateful to the personnel of the Crop Protection Committees of the 15 states we studied, for their valuable help selecting the orchards and helping with the sampling in the field.

### Funding

Hugo Enrique González-García received a scholarship from SECIHTI—Mexico for his MSc studies. This research is part of the projects: 'Distribución y rango de hospedantes del virus de la Leprosis de los cítricos, citoplasmático y nuclear, y su relación con la especie del ácaro vector: información para determinar el riesgo de dispersión del virus y generar una propuesta de manejo y contención' (Grant No. PM16-1078) funded by The Crop Protection Committee of the state of Chiapas, and 'La leprosis de los cítricos en especies agrias: daños, prevalencia y estrategia de control' (Grant No. PM18-1051) funded by The Crop Protection Committee of the state of Sinaloa. There was no additional external funding received for this study. The funders had no role in study design, data collection and analysis, decision to publish, or preparation of the manuscript.

### Grant Disclosures

The following grant information was disclosed by the authors:
The Crop Protection Committee of the state of Chiapas: PM16-1078.
The Crop Protection Committee of the state of Sinaloa: PM18-1051.
SECIHTI.

## Competing Interests

The authors declare there are no competing interests.

## Author Contributions

- Hugo Enrique González-García conceived and designed the experiments, performed the experiments, analyzed the data, prepared figures and/or tables, authored or reviewed drafts of the article, and approved the final draft.
- Ma. Teresa Santillán-Galicia conceived and designed the experiments, performed the experiments, analyzed the data, prepared figures and/or tables, authored or reviewed drafts of the article, and approved the final draft.
- Laura Delia Ortega-Arenas conceived and designed the experiments, authored or reviewed drafts of the article, and approved the final draft.
- Guadalupe Valdovinos-Ponce conceived and designed the experiments, authored or reviewed drafts of the article, and approved the final draft.
- Alberto Enrique Becerril-Román conceived and designed the experiments, authored or reviewed drafts of the article, and approved the final draft.
- Pedro Luis Robles-García conceived and designed the experiments, authored or reviewed drafts of the article, and approved the final draft.
- Ariel Wilbert Guzmán-Franco conceived and designed the experiments, analyzed the data, prepared figures and/or tables, authored or reviewed drafts of the article, and approved the final draft.
- Alfredo Sánchez-Villarreal performed the experiments, authored or reviewed drafts of the article, designed the primers for the complete RNA2 sequence of the virus, and approved the final draft.

## Field Study Permissions

The following information was supplied relating to field study approvals (*i.e.*, approving body and any reference numbers):

Fieldwork for this study was conducted under the supervision and coordination of the National Service for Agrifood Health, Safety, and Quality (SENASICA) in Mexico, as part of its Phytosanitary Epidemiological Surveillance Program. While no formal written field permit was issued specifically for this research, all sampling activities were carried out with the verbal consent of landowners and under the official authority and guidance of SENASICA personnel, in accordance with ongoing national surveillance efforts initiated by the agency in 2011.

## DNA Deposition

The following information was supplied regarding the deposition of DNA sequences:

The sequences are available at GenBank: PV533983, (CiLV-C) MT056048–MT056049, (OFV) MT073364–MT073367, (OFV-Cit1) MT073361–MT073363, (OFV-Cit2) MT073355–MT073360.

## Data Availability

Raw data is available in the Supplemental Files.

## Supplemental Information

Supplemental information for this article can be found online at http://dx.doi.org/10.7717/peerj.19889#supplemental-information.

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
