# Peer review of "Distribution and host range of viruses associated with the citrus leprosis disease complex in Mexico"

_PeerJ, doi:10.7717/peerj.19889_

## Round 0.1 · original submission · Minor Revisions

Plant viruses are significant plant pathogens that must be controlled before they cause substantial damage. Your study provides valuable insights into the citrus leprosis disease complex, including its distribution and host range. However, certain technical details should be addressed to further enhance the article. I strongly recommend carefully reviewing the reviewers' suggestions and thoughtfully considering each recommendation. If you disagree with any suggestion, it would be helpful to provide clear, well-reasoned justifications for your viewpoint.

**Language Note:** The review process has identified that the English language must be improved. PeerJ can provide language editing services - please contact us at [email protected] for pricing (be sure to provide your manuscript number and title). Alternatively, you should make your own arrangements to improve the language quality and provide details in your response letter. – PeerJ Staff

Reviewer 1 ·

Basic reporting

This manuscript presents original research on the distribution and host range of Citrus leprosis virus C (CiLV-C) and Orchid fleck virus – citrus strain (OFV-Cit) in citrus orchards across Mexico. The study addresses gap in the literature. The research is well-conceived, and the methods are appropriate for the objective outlined. The manuscript generally follows an appropriate scientific structure and provides sufficient background and citations to contextualize the research. However, there are several areas requiring revision to meet the standards of clarity, professionalism, and technical accuracy expected in scientific writing.
Language and Grammar:
There are several grammar, punctuation, and language issues throughout the manuscript that need to be addressed. Specific examples include:
• Awkward phrasing, such as in lines 31–32 (“have been done”) and 36–37 (“this effect was opposite”).
• Incorrect word usage, for example, using “despite” followed by a full clause (lines 29–30).
• Uncommon pluralization, such as “samplings,” which is rarely used in the plural form.
• Misplaced adverbs, e.g., “significantly” in line 71, which should precede “affect.”
• Incorrect scientific nomenclature, such as “Citrus × paradise,” which should be corrected to Citrus × paradisi.
• Overly long or complex sentences and lists without appropriate breaks, particularly in lines 83–85 and 90–91.
• Awkward sentence structure, such as in lines 280-282 (“…poor survival of B. yothersi is very poor on lime trees…”).
• Punctuation issues, including missing commas—for example, in lines 87–88 before the word “respectively.”
• Italicization: Italicize virus taxonomic names at the genus level and above (e.g., family, order, genus, etc.). Genus names like Cilevirus, Dichorhavirus and Higrevirus should be properly italicized in accordance with scientific conventions.
A thorough language edit is strongly recommended to ensure the manuscript meets the journal’s expectations for clarity and professional presentation.
Figures:
• Figure 3A and Figure 3B should be combined into a single bar graph with different colors representing each virus. Consolidating the data in one figure would facilitate direct comparison and improve visual clarity.
• Figures 4 and 5 should be merged into one figure with two sections: one bar graph for lime (A) and one for orange (B), using color to distinguish between viruses. This would make the information more accessible and easier to interpret.
Table:
• Although p-values are mentioned in the text, a table summarizing the full statistical analysis should be included. This would enhance transparency, support reproducibility, and align with best practices for data reporting.

Experimental design

Experimental design – clarification required before acceptance:
Before the manuscript can be accepted, several important questions related to the experimental design need to be clarified to ensure the robustness and interpretability of the results:
1. Why was the investigation limited to just 15 states, rather than covering the entire country, especially considering that the goal was to assess the nationwide distribution of viruses? Clarifying the selection criteria for these regions would strengthen the validity of the study’s conclusions.
2. What about the timing of the sampling? Was it conducted across different seasons, given that seasonal variation is important for plant- and insect-borne viruses? Failure to account for seasonal variation in sampling could have led to an underestimation of the viruses' distribution associated with the disease.
3. Was the sampling carried out simultaneously across the different geographical regions? If so, were the weather conditions comparable among these regions at the time of sampling?
Environmental factors, especially weather conditions, can significantly influence the presence and detectability of plant viruses, particularly those transmitted by insect vectors. If samples were collected at the same time across regions with differing climates or seasonal stages, it could introduce bias or inconsistencies in the observed virus distribution. Addressing this helps evaluate whether the study’s findings truly reflect spatial distribution or are confounded by temporal or environmental variation.
4. Did the experimental design include both positive and negative controls to validate the RT-PCR assays? Including this information is essential for evaluating the reliability and reproducibility of the diagnostic results.

Validity of the findings

While the study is well-structured and the conclusions are appropriately linked to the data, revisions to the Results and Discussion sections are recommended to improve clarity and readability, and to ensure key interpretations are well-supported.
Results:
The results section currently feels a bit repetitive, especially with phrases like “the greatest proportion of infected leaves was found in…” popping up multiple times, along with similar sentence structures when talking about infection levels in different states. Making these sentences more concise and varied would make the text easier to read and help the main points stand out better. Also, grouping the findings about lime and orange infections together in a more organized way could help readers follow the comparisons without getting lost.
Discussion:
1. In the discussion section, is there any evidence or reference supporting the claim made in lines 247-248 that “if a single leaf is initially infected by two viruses, one virus may eventually dominate and outcompete the other”? It’s important to support such claims with solid evidence when making an argument.
2. The statement in lines 266-268, “Furthermore, although we detected CiLV-C in lime leaves, it is likely that they are not affected in the same way as orange leaves and may not experience any negative effect—though this needs to be confirmed experimentally,” is unclear and would benefit from further clarification.

Reviewer 2 ·

Basic reporting

The paper is written in clear English, using correct technical terms. The bibliographic citations are pertinent to the topic addressed. The figures of the symptoms of leaf samples evident some differences between the viruses identified in the citrus varieties cultivated in the regions studied. The tables confirm the differences in the occurrence of the species of viruses causing leprosis in the citrus varieties evaluated. The results obtained made it possible to verify the prevalence of CiLV-C in all citrus varieties evaluated and the low prevalence of OFV/OFV-Cit strains in the same varieties. One of the possible hypotheses could be due to the presence of the vector mite B. yothersi. Even though previously obtained results also point to this fact, in order to conclude with certainty it will be necessary to carry out experiments on the transmission of viruses by the mite, for orange and lime species in the producing regions of Mexico.

Experimental design

The study has a well-defined objective and addresses a topic of great relevance to citrus farming and is in line with the scope of the journal. Leprosis needs to be controlled with acaricides to avoid causing severe economic losses to producers. The study aimed to identify the occurrence of viruses that cause leprosis in varieties grown in different producing regions of the country. The experiments were well designed, conducted and detailed, in line with the scope of the journal.

Validity of the findings

The results of the experiments carried out in this study are original, since they were a survey to identify the occurrence of different viruses that cause leprosis in varieties grown in the producing regions of Mexico. The experimental design used can be easily reproduced, since it is well-founded in well-documented literature, adding important data to the literature, such as the identification of CiLV-C in asymptomatic orange leaves and the lower transmission of CiLV-C in lime leaves by B. yothersi, compared to orange or mandarin. The statistical data was well conducted, but it is important to conclude based on this data, that the virus incidence can be multifactorial and more data are needed to infer which factors affect the prevalence of the viruses studied.

Additional comments

The study has a well-defined objective and addresses a topic of great relevance to citrus farming and is in line with the scope of the journal. Leprosis needs to be controlled with acaricides to avoid causing severe economic losses to producers. The study aimed to identify the occurrence of viruses that cause leprosis in varieties grown in different producing regions of the country. The experiments were well designed, conducted and detailed, in line with the scope of the journal.

Annotated reviews are not available for download in order to protect the identity of reviewers who chose to remain anonymous.

---

## Round 0.2 · Minor Revisions

I appreciate your constructive attitude toward the reviewers' suggestions and your efforts to improve your article accordingly. Although you have revised your manuscript based on their feedback, further improvements are still needed. Please upload a track changes file that clearly shows all modifications made in response to the reviewers' comments. Additionally, the current English revision is not at an acceptable level. I recommend seeking assistance from a colleague or using our editing service to ensure that the language is polished and professional.

**PeerJ Staff Note**: Please ensure that all review, editorial, and staff comments are addressed in a response letter and that any edits or clarifications mentioned in the letter are also inserted into the revised manuscript where appropriate.

**Language Note**: The review process has identified that the English language must be improved. PeerJ can provide language editing services - please contact us at [email protected] for pricing (be sure to provide your manuscript number and title). Alternatively, you should make your own arrangements to improve the language quality and provide details in your response letter. – PeerJ Staff

---

## Round 0.3 · accepted · Accept

I would like to thank you for accepting the referees' suggestions and improving your article based on their suggestions. Your article is ready to publish. We look forward to your next article.

Reviewer 1 ·

Basic reporting

I have reviewed the revised manuscript and confirm that the authors have addressed my previous comments regarding clarity and language. The manuscript is now written in clear, professional English.

Experimental design

The structure, references, and data presentation are appropriate for publication. I believe the manuscript meets the standards for publication.

Validity of the findings

The findings are valid.